Evidence synthesis  

physiology, ecology

glucocorticoids, meta-analysis, stress, urbanization, wildlife

**Author for correspondence:**
Maider Iglesias-Carrasco
e-mail: miglesias15@gmail.com

# Stress in the city: meta-analysis indicates no overall evidence for stress in urban vertebrates

Maider Iglesias-Carrasco, Upama Aich, Michael D. Jennions and Megan L. Head

Division of Evolution, Ecology and Genetics, Research School of Biology, Australian National University, Canberra, Australian Capital Territory, Australia

(iD) MI-C, 0000-0003-0349-7967; UA, 0000-0003-2576-0922; MDJ, 0000-0001-9221-2788; MLH, 0000-0002-8123-7661

As cities continue to grow it is increasingly important to understand the long-term responses of wildlife to urban environments. There have been increased efforts to determine whether urbanization imposes chronic stress on wild animals, but empirical evidence is mixed. Here, we conduct a meta-analysis to test whether there is, on average, a detrimental effect of urbanization based on baseline and stress-induced glucocorticoid levels of wild vertebrates. We found no effect of urbanization on glucocorticoid levels, and none of sex, season, life stage, taxon, size of the city nor methodology accounted for variation in the observed effect sizes. At face value, our results suggest that urban areas are no more stressful for wildlife than rural or non-urban areas, but we offer a few reasons why this conclusion could be premature. We propose that refining methods of data collection will improve our understanding of how urbanization affects the health and survival of wildlife.

## 1. Introduction

Human activities are altering environments worldwide, which is lowering the survival of individual wild animals, and shrinking the ranges of many species. Urbanization is one of the most drastic anthropogenic environmental changes [1]. In urban areas wildlife often encounter novel, potentially stressful perturbations. These include changes in food resources, predation pressure and new species interactions (review: [2]), as well as increased disturbance from people (e.g. [3]), and exposure to their pollutants (e.g. light and noise [4]). Some species seem to be unable to survive in urban areas and completely avoid them (urban avoiders), while others show some capacity to cope (urban adapters) [5]. A few species are even able to capitalize on environmental changes associated with urbanization and show large increases in population size as a result (urban exploiters) [6–9]. As urbanization expands it is becoming increasingly important to understand how wild animals cope with anthropogenic disturbance to predict long-term population effects and formulate appropriate conservation strategies.

In vertebrates, levels of glucocorticoid hormones, such as corticosterone, can provide information about the impact of environmental factors on individuals' fitness [10–12]. In response to an immediate stress (hereafter 'stress-induced response'), the neuroendocrine system mediates a rapid release of glucocorticoids [13,14]. This increase in circulating levels is considered to be adaptive as it mediates physiological reactions such as the suppression of the immune system and reproduction to thereby facilitate immediate behavioural responses that ensure survival [15–17]. Once the stressor abates, the neuroendocrine system then returns to its former state, ceases to induce the release of glucocorticoids and allows the individual to resume normal activities [16]. By contrast, however, when individuals experience chronic, on-going stress the functioning of the neuroendocrine system is dysregulated [16], which might result in a complex variety of responses. For example,

Proc. R. Soc. B 287: 20201754

chronically stressed individuals can show both elevated [18] and reduced [19] baseline glucocorticoid levels. Similarly, chronic stress can, in some cases, result in excessive glucocorticoid surges following exposure to an acute stressor, and a delay in the time it takes for glucocorticoid levels to return to baseline levels (review: [20]); while in other cases chronic stress results in the downregulation of the neuroendocrine system, decreasing the ability to cope with the stressor and impairing adaptive stress-induced responses (review: [12]). Both increases and reductions in glucocorticoid levels after repeated or prolonged exposure to stressors can result in the maladaptive allocation of energy to certain biological processes, resulting in a range of detrimental effects (e.g. impaired immune response [19], reduced lifetime reproduction [21,22] and lower survival [23]). If urban environments present stressful, novel challenges then wild animals could maladaptively respond to chronic stress which might, in turn, lower population growth and threaten the persistence of wild populations in urban areas.

To understand the physiological responses of wildlife to urbanization, studies often compare baseline and stress-induced glucocorticoid levels (i.e. the response to a sudden, temporarily stressful event such as a predatory attack) between urban and non-urban populations (table 1). It is generally assumed that urban populations will show higher baseline levels of glucocorticoids due to chronic stress, but the results are mixed: while some studies find higher baseline glucocorticoid levels in urban populations [26,37], others find no effect [24,36] or even the opposite effect [30,45]. Similarly, the stress-induced responses of chronically stressed urban dwellers is generally expected to be higher than that of their non-urban counterparts, but, in this case too, the evidence is mixed [14,25,36]. This variation among studies could be related to the complex responses of the neuroendocrine system to chronic stress (highlighted above), which could constrain the physiological capacity to respond to the stressor. However, the variation could also arise from species or population/or individual-level differences in the perception of the stressor, and hence from intrinsic factors associated with the individuals being measured (e.g. their reproductive status [56], sex [57] or taxa [58]); from variation between different urban environments; or from methodological factors, such as the source of glucocorticoids (e.g. feather/hair versus blood) that might reflect cumulative versus current levels of stress (e.g. [42,53]). To date, however, the role of these various potentially moderating factors has not been formally examined.

Here, we use a meta-analysis of 34 studies from 27 species for either baseline or stress-induced glucocorticoid levels to: (i) test whether there is, on average, evidence for greater chronic stress of wild animals in urban environments as revealed by either baseline glucocorticoid levels, or the response to experimentally induced stressors; and (ii) estimate the importance of four biological factors (sex, life stage, breeding season, taxa), an environmental factor (human population size as proxy for city size) and a methodological factor (source of hormone sample) that might account for variation in the outcome of different studies. We then discuss the implications of our results for the conservation of wildlife in urban areas.

## 2. Material and methods

### (a) Article search
We performed a search to identify published articles that test the effect of urbanization on glucocorticoid levels of wild animals.

We conducted a topic search in Web of Science (day of search 29 April 2019) by including the terms 'urban* OR cities OR city OR anthrop*' AND 'corticosterone OR cortisol OR "cort" OR glucocorticoid*'. We also conducted the same search in Scopus, but here we restricted our search to the subject areas of Environmental Sciences and Agricultural and Biological Sciences. This resulted in a pool of 1400 papers in Web of Science and 429 in Scopus. Of these 396 were duplicates, giving us a total of 1433 studies. Once we had screened and collected data from these papers we ran a second search (day of search 20 February 2020) using the same criteria, but restricted to papers published since the date of our first search. This added another 16 papers, for a final tally of 1449 papers.

### (b) Inclusion criteria
We used the software Rayyan [59] to screen the titles and abstracts of the 1449 papers. We identified empirical studies that measured the stress response of wild animals across habitats that differed in the level of urbanization. After this first round of filtering, we retained 137 papers. We then read the methods of these papers to ensure that they met the criteria for inclusion in our meta-analysis. These criteria were:

(1) The study had information on equivalent measures of glucocorticoid levels from individuals from both an urban and at least one non-urban population, or from individuals sampled along an urbanization gradient with data available for the most extreme values along the gradient (e.g. urban and non-urban sites). We discarded the studies or locations corresponding to suburban areas.
(2) The study was on wildlife, not humans, nor domesticated or captive animals (e.g. caged or in zoos).
(3) The study measured the glucocorticoid levels of wild captured animals in urban and non-urban populations, and not a response to an indirect measure of urbanization (i.e. light pollution), or a response to an added chronic stress (e.g. keeping animals caged before taking measures for long periods of time).
(4) The study measured naturally occurring rather than experimentally manipulated glucocorticoid levels by injection of these hormones.

After assessing the methods based on these inclusion criteria we ended up with 34 articles (see electronic supplementary material, figure S1 for a PRISMA diagram). All searches and assessment of inclusion criteria were conducted by M.I.-C. and U.A.

### (c) Data extraction
For the 34 papers that met our selection criteria for data extraction, we recorded the mean, standard deviation and sample size for baseline and stress-induced (when available) glucocorticoid levels for populations from both urban and non-urban habitats. Baseline glucocorticoid levels were considered as those measured from cumulative sources (e.g. feather, hair) or current levels taken within 3 min of the capture of animals for blood. Stress-induced glucocorticoids were those measures taken after a short stressful event. In the case of birds, stress-induced hormones were measured from blood after handling and restraint, usually after holding the individual in a cloth bag for 10–60 min. A similar approach was used by the only study included on fish [43]. In mammals, stress-induced measures were taken from faecal samples after capture–recapture [52] or in animals kept in traps for several hours [47]. The single study that measured stress-induced glucocorticoids in an amphibian collected samples after an agitation test, which involved regular agitation of the container in which the animals were housed (agitation for 1 min every 3 min for 1 h) [40].

**Table 1.** Studies used in this meta-analysis that document the glucocorticoid levels of vertebrates from urban and non-urban habitats.

| reference | taxa | species | type of measure | country |
|---|---|---|---|---|
| Abolins-Abols et al. [24] | bird | Junco hyemalis | baseline/stress-induced blood corticosterone | USA |
| Atwell et al. [25] | bird | Junco hyemalis | baseline/stress-induced blood corticosterone | USA |
| Beaugeard et al. [26] | bird | Passer domesticus | baseline feather corticosterone | France |
| Beck et al. [27] | bird | Melospiza melodia | baseline/stress-induced blood corticosterone | USA |
| Bonier et al. [28] | bird | Zonotrichia leucophrys | baseline blood corticosterone | USA |
| Brunton et al. [29] | mammal | Macropus giganteus | baseline faecal cortisol | Australia |
| Buxton et al. [30] | bird | Agelaius phoeniceus | baseline faecal corticosterone | USA |
| Chávez-Zichinelli et al. [31] | bird | Columbina inca, Melozone fusca | baseline faecal corticosterone | Mexico |
| Corbel et al. [32] | bird | Columba livia | baseline/stress-induced blood corticosterone | France |
| Davies et al. [14] | bird | Pipilo aberti | baseline/stress-induced blood corticosterone | USA |
| Davies et al. [33] | bird | Pipilo aberti | baseline/stress-induced blood corticosterone | USA |
| Davies et al. [34] | bird | Troglodytes aedon | baseline/stress-induced free blood corticosterone | USA |
| Davies et al. [35] | bird | Melospiza melodia | baseline blood corticosterone | USA |
| Fokidis & Deviche [18] | bird | Toxostoma curvirostre | baseline blood corticosterone | USA |
| Fokidis et al. [36] | bird | Toxostoma curvirostre, Mimus polyglottos, Pipilo aberti, Passer domesticus | baseline/stress-induced free and total blood corticosterone | USA |
| Fokidis et al. [37] | bird | Toxostoma curvirostre | baseline/stress-induced blood corticosterone | USA |
| Foltz et al. [38] | bird | Melospiza melodia | baseline/stress-induced blood corticosterone | USA |
| French et al. [39] | reptile | Urosaurus ornatus | baseline/stress-induced blood corticosterone | USA |
| Gabor et al. [40] | amphibian | Eurycea tonkaway | baseline/stress-induced water corticosterone | USA |
| Grunst et al. [41] | bird | Melospiza melodia | baseline/stress-induced blood corticosterone | USA |
| Ibáñez-Álamo et al. [42] | bird | Turdus merula | baseline feather corticosterone | Spain, France, Finland |
| King et al. [43] | fish | Micropterus salmoides | baseline/stress-induced blood cortisol | Canada |
| Łopucki et al. [44] | mammal | Apodemus agrarius | baseline faecal corticosterone | Poland |
| Lyons et al. [45] | mammal | Tamias striatus | baseline faecal/hair cortisol | Canada |
| Meillère et al. [46] | bird | Passer domesticus | baseline/stress-induced blood corticosterone | France |
| Nelson et al. [47] | mammal | Vulpes macrotis | baseline/stress-induced faecal cortisol | USA |
| Parry-Jones et al. [48] | mammal | Pteropus poliocephalus | baseline faecal corticosterone | Australia |
| Partecke et al. [49] | bird | Turdus merula | baseline/stress-induced blood corticosterone | Germany |
| Rebolo-Ifran et al. [50] | bird | Athene cunicularia | baseline feather corticosterone | Argentina |
| Scheun et al. [51] | mammal | Galago moholi | baseline faecal cortisol | South Africa |
| Shimamoto et al. [52] | mammal | Sciurus vulgaris | baseline/stress-induced faecal cortisol | Japan |
| Stothart et al. [53] | mammal | Sciurus carolinensis | baseline faecal/hair cortisol | Canada |
| Wright & Fokidis [54] | bird | Cardinalis cardinalis | baseline/stress-induced blood corticosterone | USA |
| Zhang et al. [55] | bird | Passer montanus | baseline blood corticosterone | China |

We relied on author descriptions to categorize sites as urban or non-urban habitats. Although the exact type of city varied considerably across the studies, in general, urban individuals were sourced from green spaces within the metropolitan area, such as city parks, industrial areas or school campuses. Non-urban areas were defined as nature reserves, natural forests, grasslands or even crops, depending on the study. To explore whether and to what extent variability between urban areas affected the response in glucocorticoids, we recorded the human population size of each city as a proxy for its degree of urbanization (data mainly from government entities such as Eurostat, United States Census Bureau; see raw data available from the Dryad digital Repository: https://doi.org/10.5061/dryad.pzgmsbcj5 [60]).

When the studies did not provide data on glucocorticoids in the text, we calculated the mean and standard deviation (or error) from the raw data (if supplied) or from figures using the *metaDigitise* [61] package in R. If a study reported glucocorticoid values for different years or different cities, we recorded each separately. When available, we recorded separate values for males and females, juveniles and adults, different seasons (breeding, non-breeding or moulting), each species, and the sample from which hormones were measured (feather, faeces, hair, water

or blood-separating free—unbound to corticosteroid-binding globulin—from total—bound and unbound to corticosteroid-binding globulin—blood corticosterone). These variables, as well as the human population size of each city, were later used as moderating variables in our models (see below).

The data from each paper was extracted independently by M.I.-C. and U.A. When values measured from figures were similar we used the average, but when there were large discrepancies between the two measures ($n = 3$) due to measurement error, we took a third measure, and we used the mean of the two most similar measures in our analysis.

## (d) Calculating effect sizes

We used the standardized mean difference, Hedges's $g$ [62] as our effect size for the difference in glucocorticoid levels between urban and non-urban populations. Negative effects represent lower glucocorticoid levels in the urban population. When a study measured glucocorticoid values for the same group of individuals (e.g. multiple years or from a different sampling source) more than once, we calculated an effect size for each and controlled for non-independence of the data statistically [63] (see below). When a study reported stress-induced glucocorticoid levels at multiple time intervals after exposure (papers [25,32,49]), we used the maximum levels of glucocorticoids, since most other studies made their measurements at the known peak response for the focal species. We obtained a total of 108 effect sizes for baseline, and 54 for stress-induced glucocorticoids. The sample sizes were, however, smaller for analyses that included moderators (i.e. sex, season, life stage, source of hormone and taxa) as we excluded levels that had small sample sizes (i.e. less than or equal to 2 studies or less than or equal to 2 species, and less than or equal to 6 effect sizes; see electronic supplementary material, table S1 for detailed information on exclusion criteria).

## (e) Statistical analysis

We ran our analyses using the package *metafor* [64] and the function rma.rm in R v. 3.5.1. We conducted separate analyses for baseline and stress-induced glucocorticoids. For both, we conducted a multi-level meta-analysis of birds only (baseline: 16 of the 27 species; stress-induced: 10 of 15), as well as one including all taxa. Next, we conducted meta-regressions to look at the effects of our moderating variables. Study identity, group identity nested within the study, and species identity were included as random effects. Both study identity and species identity were included because some studies had data for multiple species and some species occurred in multiple studies, and so there was potential for each to explain variance that the other did not. All models also included an observation level random effect to estimate the residual variance [65]. In general, model design and data exclusion was based around ensuring adequate sample sizes for each level of any included moderator.

### (i) Testing whether to include the phylogeny

To quantify the amount of heterogeneity explained by phylogeny, and whether it was appropriate to include phylogeny in the 'bird only' models, we first ran a meta-analysis in which we controlled for phylogenetic dependence by adding a correlation matrix of species relatedness to the species-level random effect. We computed the least-squares consensus tree from a random sample of 1000 trees from published data [66]. This analysis indicated that phylogeny explained little to no additional heterogeneity ($I^2 = 0.00$; see below for method used to estimate $I^2$) and that adding phylogeny gave a worse model fit (i.e. higher AICc). For these reasons, we did not include the phylogeny in our 'birds only' analysis.

Similarly, we did not control for phylogeny in our analyses of all taxa for three reasons: the imbalanced nature of our dataset (i.e. few non-birds); the lack of heterogeneity explained by the phylogeny in the bird-only model and the difficulty in obtaining fully resolved phylogenetic tree topologies that include diverse taxa [67,68].

### (ii) Baseline glucocorticoid levels

Birds only: first, we ran a meta-analysis to determine the mean effect size. To quantify the heterogeneity attributable to our random effects (i.e. study, group within study and species), we quantified the heterogeneity statistic $I^2$ [69,70] using the $I^2$ function in the package metaAidR (link in 'daniel1noble/metaAidR'). We then ran four separate meta-regressions to determine the moderating effect of source of hormone sample (total/free blood, feather or faeces), sex (males, females), season (breeding, non-breeding) and human population size (log-transformed) for adults only.

All taxa: we repeated the general approach described above but including all taxa. First, we ran a meta-analysis to determine the mean effect size across the entire dataset. From this model, we quantified the heterogeneity explained by each of our random effects. We then conducted six separate meta-regressions (excluding amphibians, reptiles, fish and hormones sourced from water due to the small sample sizes of these factor levels, see electronic supplementary material, table S1) to look at the moderating effects of taxa (bird, mammal), season (breeding, non-breeding), life stage (juvenile versus adult), sex (male, female), source of hormone sample (feather, hair, total/free blood or faeces) and log-transformed human population size of each city. Here, the meta-regressions looking at the effects of season and sex controlled for potential broad-scale taxonomic effects by including taxa, but the source of the hormone sample did not because taxa and hormone source were highly correlated (e.g. feathers are only measured in birds and hair only measured in mammals).

### (iii) Stress-induced glucocorticoid levels

Due to the high representation of birds in our dataset (48 of 54 effects), we conducted a meta-analysis of stress-induced glucocorticoid levels for birds on their own as well as for all taxa combined. For both datasets, we also ran a meta-regression that included log-transformed human population size as moderator. We then conducted two meta-regressions to look at the moderating effects of season (breeding, non-breeding) and hormone source (free blood corticosterone, total blood corticosterone) in adult male birds. Females and juveniles were excluded from these analyses due to their small representation in the dataset (electronic supplementary material, table S1).

### (iv) Sensitivity analyses

There was one obvious outlier for each baseline and stress-induced glucocorticoids (electronic supplementary material, figure S2). Rerunning the models without these data points did not influence our conclusions (electronic supplementary material, tables S2–S4). In addition, some studies contributed more than one observation, for example, by repeatedly measuring the same individuals (statistically controlled for in the models outlined above by including the random term 'group identity'). To test the robustness of our results, and as an alternative control for non-independence, we used a single effect size for each group. To do this we randomly selected one effect size per group and reran our analyses. In no case did these models change our main conclusions (shown only for meta-analytic models in electronic supplementary material, tables S2–S4). Finally, since significant results are more likely to be published, we tested for publication bias. First, using funnel plots, we looked for asymmetries in the relationship between the meta-analytic residuals and the inverse of their precision (i.e. the

(a)  baseline glucocorticoids

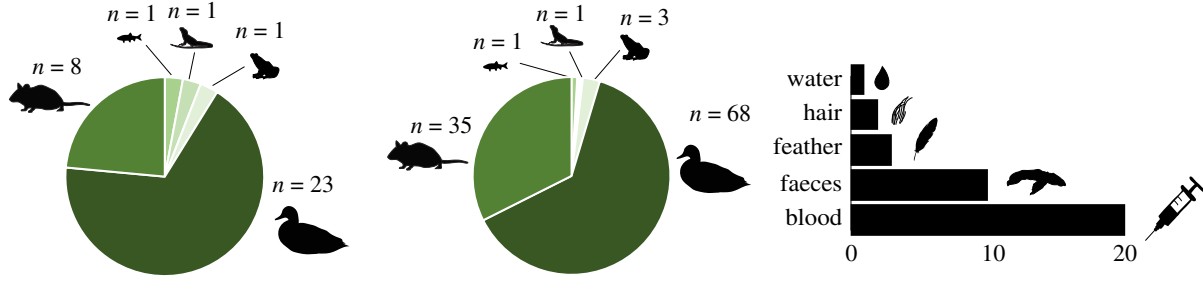

(b)  stress-induced glucocorticoids

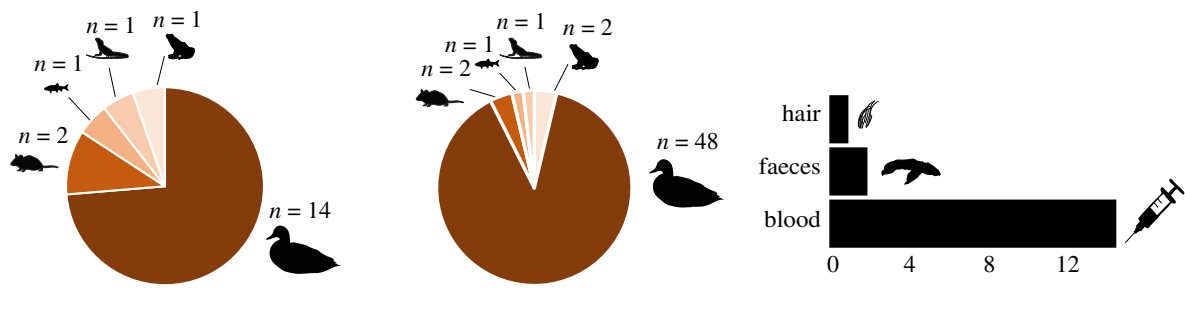

**Figure 1.** Pie charts show the number of studies and the number of effect sizes for (a) baseline and (b) stress-induced glucocorticoid levels for each taxa in the meta-analysis. Bar plots show the number of studies for each source of hormone samples. (Online version in colour.)

inverse standard error of the effect size, $1/SE_i$, electronic supplementary material, figure S2). Second, we ran Egger's regressions by adding the precision term as a moderator to each meta-analysis of baseline and stress-induced glucocorticoids [70], and tested the significance of the slope and intercept.

## 3. Results

From the 34 studies that met our inclusion criteria (table 1), there were 108 effects from 27 species for baseline glucocorticoids and 54 effects from 15 species and 19 studies for stress-induced glucocorticoids (figure 1).

### (a) Baseline glucocorticoids

The average effect of urban habitat on baseline glucocorticoid levels was negligible, both for birds alone (mean $g = 0.105$, 95% CI = −0.224–0.435, $n = 68$ effects; electronic supplementary material, table S2), and for all vertebrates (mean $g = 0.068$, 95% CI = −0.135–0.272, $n = 108$ effects; electronic supplementary material, table S3). For birds, species identity explained a moderate proportion of the heterogeneity in effect sizes ($I^2 = 0.395$, 95% CI = 0.222–0.562), while the variation explained by study identity and the group (nested within the study) was negligible. For all taxa combined, species identity explained a small proportion of the variation ($I^2 = 0.152$, 95% CI = 0.089–0.233), while the amount explained by study and group identity was negligible (see electronic supplementary material, tables S2 and S3).

We did not find any moderating effect of sex, source of hormone sample, season, taxa or human population size in our bird-only analysis (electronic supplementary material,

table S2 and figure S3). We also found no moderating effects of these variables or life stage in our analysis of all taxa combined (electronic supplementary material, table S3; figure 2). Sensitivity analyses showed that neither the removal of an outlier, nor the way we controlled for group identity affected the results (electronic supplementary material, tables S2 and S3).

### (b) Stress-induced glucocorticoids

The average effect of urban habitat on stress-induced glucocorticoid levels was negligible both for birds alone (mean $g = -0.209$, 95% CI = −0.530–0.113, $n = 48$ effects; electronic supplementary material, table S4) and for all vertebrates (mean $g = -0.079$, 95% CI = −0.359–0.202, $n = 54$ effects; electronic supplementary material, table S4). The heterogeneity in effect sizes explained by the species, group and study identities was negligible (electronic supplementary material, table S4). There was no detectable effect of human population size in the dataset for all taxa combined or that for birds only, and neither season or the source of hormone explained variation in the effect sizes for male birds only (electronic supplementary material, table S4; figure 2)

Finally, we did not detect any publication bias for either baseline (slope = −0.071, $p = 0.345$, 95% CI = −0.219–0.078) or stress-induced glucocorticoids (slope = −0.195, $p = 0.164$, 95% CI = −0.473–0.083) (electronic supplementary material, figure S2).

## 4. Discussion

We found no overall difference in either baseline or stress-induced glucocorticoid levels in wild animals between urban

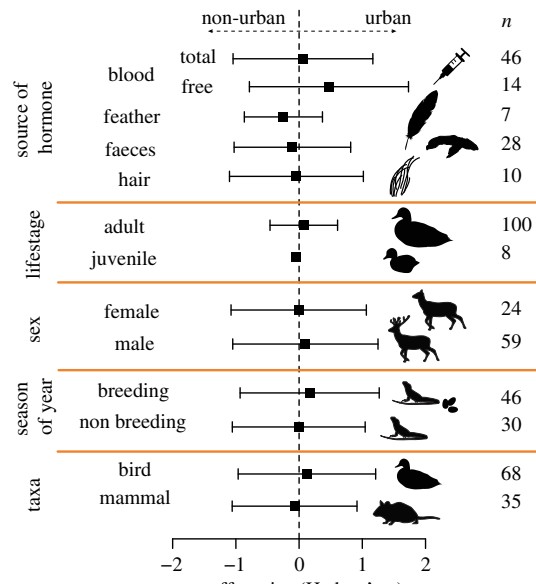

**Figure 2.** (a) Response to being in an urban habitat of baseline glucocorticoid levels of all taxa (mean effect size and 95% confidence intervals), and partitioned on the basis of five moderators (source of hormone sample, life stage, sex, season and taxa). (b) Response to being in an urban habitat of stress-induced glucocorticoid levels of all taxa, all birds and male birds partitioned by season and source of hormone. Positive effect sizes indicate higher glucocorticoid levels in urban habitats. Numbers represent the sample size for effect size. (Online version in colour.)

and non-urban habitats. This finding contrasts with the general expectation that anthropogenic disturbance, specifically urbanization, causes chronic stress and, hence, alters levels of glucocorticoids. It is, of course, possible that urban areas are no more stressful for wildlife than non-urban areas, at least for those vertebrates that are capable of living in urban areas. Stress imposed by novel ecological pressures in urban habitats, such as noise or light pollution, might have a comparable physiological effect to that imposed by stressors in natural habitats, such as a greater abundance of predators or constant searching for food. Another possibility is that the lack of a detectable stress response to urbanization arises from looking at urban areas *per se*, rather than specific anthropogenic pressures, such as pollution or diet quality [71]. For example, a meta-analysis on fishes showed that anthropogenic noise can have negative effects on their behaviour and physiology, but the fitness consequences depended on the noise source [72]. Similarly, a recent comparative analysis found that noise, but not light, pollution was negatively associated with baseline corticosterone levels in birds [73]. The generic use of 'urban area' as a proxy for

the presence of anthropogenic stress might be too crude an index if it includes cities that vary in their extent of urbanization (e.g. greenness versus built areas), where key stressors differ in intensity. We attempted to control for some of the variation among urban sites by including the city's human population size as a moderator, but we found no effect of city size. However, we cannot rule out that variation in other characteristics of urban areas could explain differences in the strength of the response between studies.

None of sex, breeding season, life stage nor taxa moderated the effect of urbanization on glucocorticoid levels. This is, perhaps, unsurprising. Although some intrinsic characteristics of individuals are known to strongly influence glucocorticoid production, our analysis would only detect an effect of these moderators if all species responded in the same direction. However, evidence from the literature suggests that species-specific responses to stressors are commonplace, and might involve complex interactions between several factors. For example, previous studies have shown a stronger male than female stress response to anthropogenic disturbances in some species (e.g. [74]), and the reverse pattern in others (e.g. [75]). Similarly, stress responses to environmental challenges can vary seasonally with the reproductive status of individuals. For instance, males and non-breeding female maned wolf (*Chrysocyon brachyurus*) increased their glucocorticoid levels when exposed to humans, but reproductively active females did not [76]. Testing how single biological factors moderate the effects of urbanization independently, as we did in our meta-regressions due to limited sample sizes, does not allow us to detect such interactions. Another potential reason for the lack of significant effects of our moderators is potential differences in phenology (arising from differences in seasonal or circadian cues) between urban and non-urban populations. Although the studies included in our analysis controlled for time of year when sampling, and we also considered season in our analysis, other studies have shown that urban populations and their non-urban counterparts can differ in the onset of the breeding season [77,78] and circadian rhythms [79]. Differences in stress hormone production that are related to the onset of breeding or daily activity might, therefore, mask differences in stress hormone levels that are associated with urban and non-urban habitats.

Our analysis shows that species identity explains a significant portion of the heterogeneity in the effect of urbanization on glucocorticoid levels. Although our meta-analyses suggest that urban areas are not generally stressful for birds or mammals, they also reveal the response to urbanization is highly species-specific, which implies that some species might be negatively affected. A key challenge is to determine what aspects of a species's biology or ecology predict their ability to persist in the face of urbanization. For instance, are taxa with lower mobility or higher sensitivity to chemicals (e.g. amphibians) impacted more by urbanization? One constraint of our dataset is that we could only investigate the stress response of species that occur in cities (i.e. urban exploiters). The fact that they survive in cities is, of course, likely to mean that they find urban areas less stressful than species which are absent. For instance, urban exploiters could habituate or acclimate such that a repeated stressor found in urban areas, like the presence of humans, is eventually treated as harmless [15]. However, this might not be the case for urban avoiders, explaining their inability to survive in cities. Monitoring the stress responses of species that tend to disappear from newly

urbanized areas would provide an interesting insight into the extent to which this physiological response predicts a species's persistence or disappearance [72].

It is worth noting that we cannot conclude that urban areas are no more stressful than non-urban areas based on differences in glucocorticoid levels. This is because chronic stress might change the underlying functioning of the neuroendocrine system in a variety of ways in different species and populations. For example, while stress-induced responses can increase in chronically stressed individuals (e.g. [36]), repeated exposure to stressors can also decrease the release of glucocorticoid hormones even when an individual identifies it as a threat [80]. In such conditions, stressors can eventually incur costs because they hinder an individual's ability to respond to other stressors [81]. Our results do not support a consistent response to chronic stress across species or populations, as urban and non-urban populations did not differ in the levels of either baseline or stress-induced glucocorticoids. However, this does not rule out the possibility that different populations are responding to urban stressors in opposite directions, thereby masking any overall effects. Before we can make robust recommendations in the context of urban conservation, more research is necessary to accurately assess the physiological effects of urbanization and the potential role of glucocorticoids in facilitating urban exploitation.

Hormone sampling methodology might also account for the lack of a detectable effect of urbanization on wildlife glucocorticoid levels. Some sampling sources, such as blood and saliva, only provide an opportunity to measure short-term stress responses. This is a limitation when assessing chronic stress, since prolonged high levels of glucocorticoids are only detectable when using integrated measures such as faeces, feather or hair [82]. These sources integrate levels of blood glucocorticoids secreted, metabolized and excreted by individuals over long periods of time [58]. In addition, sampling these sources is often less invasive, which lowers the stress caused by handling animals. Despite these potential advantages, few studies sampled hormones from sources that allow for these integrated measures (table 1 and figure 1). Also, in our meta-analysis most studies collected total (bound and unbound to corticosteroid-binding globulin) rather than free blood corticosterone (unbound to corticosteroid-binding globulin [83]. These two measures respond differently to stress (e.g. [84]). For instance, exposure to stress can decrease levels of corticosteroid-binding globulin and increase free glucocorticoids, while total glucocorticoids remain the same [85]. This might also account for the lack of a detectable stress response to urban areas in our meta-analysis.

Another difficulty when studying stress responses of wildlife to urbanization is the potential for individuals to move between habitats—especially in species that migrate or have large home-ranges. Both baseline and stress-induced responses might vary depending on whether the individuals measured developed within the city or migrated there as adults. For instance, stressful conditions during development could permanently alter the neuroendocrine system with a consequent increase in the stress-induced response later in life [26]. Whereas, experiencing any stress associated with urban habitats only during adulthood could lead to either increases or decreases in the baseline and stress-induced responses due to the plasticity of the neuroendocrine system [12,20]. Similarly, individuals might regularly move between urban and non-urban sites. In this case, cumulative measures, especially those from hair and feathers, might not accurately reflect the stress experienced by individuals while in urban environments as feathers and hair represent glucocorticoid levels accumulated over weeks or months [58]. In such cases, current measures of glucocorticoid levels such as those from blood, could more accurately represent the stress levels of individuals from the site at which they were collected. To explore whether the use of all sources of hormones biased our results due to the movement of individuals between habitats, we performed a *post hoc* analysis of baseline glucocorticoids using only blood samples. We did not find any significant effect of urbanization ($g = -0.002$, 95% CI $= -0.322$–$0.318$; electronic supplementary material, table S5), suggesting that the inclusion of cumulative measures does not explain the observed lack of effect in our meta-analysis. The complexity of potential hormonal responses to stress might hamper our ability to detect any directionality in the response to urbanization due to insufficient information about the origin of individuals. We suggest that empirical studies that investigate the effects of urbanization in species with high motility need to record information on the history of the individuals sampled whenever this is possible to provide a better insight into the stress responses of wildlife to urbanization.

Most published studies rely on glucocorticoids to measure the stress response to environmental disturbances due to their effects on metabolism, behaviour and the mediation of energetic demands [86–88]. However, glucocorticoid levels provide limited information about how exposure to chronic stress ultimately affects an individual's fitness [89,90]. They are only one component of a suite of physiological and behavioural responses to stress [91]. For example, chronic stress increases glucose levels [92] and oxidative stress [93], and reduces immune responses [94,95], body condition [12], and reproductive output [21,22]. Each of these responses could have negative consequences for an individual's reproductive output and, by extension, population viability. Due to the numerous physiological responses to stress, the use of glucocorticoids as a biological marker of a decline in fitness is likely to be limited and context-dependent. To predict how urban habitats impact wildlife we need to measure not only glucocorticoids but other downstream measures of stress [83].

A final point is that we cannot assume that all differences in stress response between urban and non-urban populations are due to differences in chronic stress alone. It is possible that the intrinsic characteristics of the individuals that occupy each habitat might contribute to differences in stress levels. For instance, as in the case for invasive species [96,97], it is possible that individuals who colonize urban areas, have a different phenotype to those that do not. For example, individuals at invasion fronts are more likely to show bold behaviours (e.g. explore more, more risk prone, reviewed in [2]) and some of these behaviours are known to be mediated by corticosterone. Individuals at invasion fronts, therefore, tend to have lower baseline but higher stress-induced responses than those at the core population [96,97]. Similarly, if bolder individuals are more likely to colonize urban areas the same scenario may arise in the case of wildlife responses to urbanization.

## 5. Conclusion

Given the rapid increase in human disturbance of natural habitats, research on the effects it has on the physiology of wildlife is

critical to inform conservation plans. At face value our meta-analysis offers no support for the claim that urban environments are stressful and alter glucocorticoid levels in wild animals. We have, however, highlighted some potential methodological and sampling problems with this conclusion. To move the field forward future studies should sample downstream physiological measures (e.g. immune responses), and include integrative measures of glucocorticoids (e.g. from faeces, hair and feathers). Ideally, research should focus on long-term studies where the movement, origin and reproductive history of animals is known. We should also monitor a wider range of taxa, as comparative analyses will help biologists identify which aspects of a species's biology can predict responses to urbanization. Monitoring species that rapidly disappear from newly established urban areas is especially important. Finally, we suggest that comparing the impacts of urbanization with those of other anthropogenic disturbances (e.g. exotic plantations, croplands) offers an interesting research avenue to provide a broader understanding of the physiological effects of human disturbance on wildlife.

Data accessibility. Data available from the Dryad Digital Repository: https://doi.org/10.5061/dryad.pzgmsbcj5 [60].

Authors' contributions. M.I.-C. and M.L.H. conceived the idea and designed the study; M.I.C. and U.A. collected the data; M.I.C. analysed the data with input from M.D.J. and M.L.H. M.I.C. wrote the first draft of the manuscript; all authors contributed substantially to the manuscript.

Competing interests. We declare we have no competing interests.

Funding. Funding was provided by the Australian Research Council (grant nos. DP160100285 to M.D.J. and FT160100149 to M.L.H.)

Acknowledgements. We thank Lauren Harrison, Dan Noble and Rose O'Dea for advice with the analysis, Juan Diego Ibáñez-Álamo, Jesko Partecke, Adrien Frantz and Hélène Corbel for providing necessary information about their studies and two anonymous reviewers for their comments that improved our manuscript.

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
