## [Reviewer comments · Proceedings of the Royal Society B: Biological Sciences]

Review History

RSPB-2020-1754.R0 (Original submission)

Review form: Reviewer 1

Recommendation

Major revision is needed (please make suggestions in comments)

Scientific importance: Is the manuscript an original and important contribution to its field?

Good

General interest: Is the paper of sufficient general interest?

Excellent

Quality of the paper: Is the overall quality of the paper suitable?

Good

Is the length of the paper justified?

Yes

Should the paper be seen by a specialist statistical reviewer?

Yes

Do you have any concerns about statistical analyses in this paper? If so, please specify them explicitly in your report.

No

It is a condition of publication that authors make their supporting data, code and materials available - either as supplementary material or hosted in an external repository. Please rate, if applicable, the supporting data on the following criteria.

Is it accessible?

Yes

Is it clear?

Yes

Is it adequate?

Yes

Do you have any ethical concerns with this paper?

No

Comments to the Author

Dear authors,

thank you for contributing this well written and important paper. The topic has been well researched in the last 20 years, but only reviews and opinion pieces have been published that tried to summarise the existing evidence, so I really think that a meta-analysis is now timely.

I have no major comments on the procedures you have used for your meta-analysis, although I am not an expert of these methods so I am hoping that the other referees will be able to comment more in detail on such aspects.

My only major comment refers to the definition (or better, lack of definition) of what "urban" is in your analysis. I agree with you that species-specific responses to urbanisation are likely to drive much of the variation in the data available on this topic, as your results clearly show. However, I also strongly think that another reason where this variation might arise from is the great heterogeneity of the urban environment, as well as of the difference between urban and natural environments where the animals part of these studies were measured. You sort of recognise this aspect in your discussion (L255-257), but I think there is a scope to actually quantify such heterogeneity in the dataset and account for it in your models. For instance, it should be straightforward to obtain information about where the animals were sampled (exact coordinates) from the methods section of each paper, and then use such coordinates to quantify the level of urbanisation and/or "greenness" at each of these sites, for instance using published software (http://evolecol.hu/wp-content/uploads/2016/05/Seress_etal_2014_LUP.pdf). I think this will add a lot to your paper by greatly enhancing the interpretation of the results, thereby increasing the impact of your study. I don't think it will take a lot of time to do so since we are only talking about 34 results (68 estimates of urban/rural, which should be pretty fast to obtain using a software).

Another comment refers to when such measurements are taken. You distinguish this by using the variable "season" in your dataset. However, urbanised populations might differ in when they breed compared to rural counterparts, up to even a month (Partecke et al 2004 Proc B, Dominoni et al 2013 Proc B). Since we know that cort can vary on a seasonal basis, these samples might have been taken at different points of such annual cycle for urban and rural populations. There is nothing you can do about this now, but perhaps you could write one or two sentences in the discussion about it? Similarly, there is also a diel cycle in cort, but urban and rural animals can also differ in how they partition their activity between day and night along urban gradients, for

instance mammals become more nocturnal when human activity is high (Gaynor et al 2017 Science). Again, if animals are sampled at the same time between urban and rural areas, these samples might have been taken at different times of their diel cycle in cort. So this could also be recognised in your discussion.

Thanks and I hope these comments will be helpful.

Review form: Reviewer 2

Recommendation

Accept with minor revision (please list in comments)

Scientific importance: Is the manuscript an original and important contribution to its field?

Good

General interest: Is the paper of sufficient general interest?

Excellent

Quality of the paper: Is the overall quality of the paper suitable?

Good

Is the length of the paper justified?

Yes

Should the paper be seen by a specialist statistical reviewer?

No

Do you have any concerns about statistical analyses in this paper? If so, please specify them explicitly in your report.

No

It is a condition of publication that authors make their supporting data, code and materials available - either as supplementary material or hosted in an external repository. Please rate, if applicable, the supporting data on the following criteria.

Is it accessible?

Yes

Is it clear?

N/A

Is it adequate?

N/A

Do you have any ethical concerns with this paper?

No

Comments to the Author

See attached file. (See Appendix A)

Decision letter (RSPB-2020-1754.R0)

06-Aug-2020

Dear Miss Iglesias-Carrasco:

Your manuscript has now been peer reviewed and the reviews have been assessed by an Associate Editor. The reviewers' comments (not including confidential comments to the Editor) and the comments from the Associate Editor are included at the end of this email for your reference. As you will see, the reviewers and the Editors have raised some concerns with your manuscript and we would like to invite you to revise your manuscript to address them.

Research ethics:

Use of animals and field studies:

It is a condition of publication that you make available the data and research materials supporting the results in the article. Please see our Data Sharing Policies (<https://royalsociety.org/journals/authors/author-guidelines/#data>). Datasets should be deposited in an appropriate publicly available repository and details of the associated accession number, link or DOI to the datasets must be included in the Data Accessibility section of the

article (<https://royalsociety.org/journals/ethics-policies/data-sharing-mining/>). Reference(s) to datasets should also be included in the reference list of the article with DOIs (where available).

Please submit a copy of your revised paper within three weeks. If we do not hear from you within this time your manuscript will be rejected. If you are unable to meet this deadline please let us know as soon as possible, as we may be able to grant a short extension.

Best wishes,
The Proceedings B Team
mailto: proceedingsb@royalsociety.org

Associate Editor

Board Member: 1

Comments to Author:

Thank you for submitting your manuscript for consideration as an Evidence Synthesis article in PRSB. Your work has now been seen by 2 experts in the field, and I have read the manuscript myself. I am pleased to say that there is a consensus that your article is interesting, highly pertinent for evidence synthesis, and provides a considered and objective overview of key literature and ideas. As such, I encourage you to revise the manuscript, in line with the constructive and detailed suggestions below. Overall (though please do view the criteria reproduced below) I can say that I am satisfied with the level of transparency and statistical rigour in your meta-analysis, especially the explicit inclusion of criteria for choice of data/literature, and the associated statistical and sensitivity analyses. Among the various key issues raised, is the need to provide a clearer and standardised definition of what constitutes an urban environment, and I would encourage you to consider inclusion of spatial coordinates, and the referee proposes some appropriate software, to provide a quantitative comparator of specific

sample sites. I appreciate that this may require an additional workload, but my view is that it will provide a more informed and consistent classification in relation to the inferences drawn. I also emphasise the need to provide some consideration of the likely impact of when samples were collected, and whether any seasonal variation in reproductive cycles for example, are likely to have an impact. There is also a need to provide a more explicit, though brief, consideration of the complexity of stress-induced endocrine responses, as well as a more considered approach of the likely impacts of using a diversity of sample types, encompassing feathers, hair, blood and faeces. It is also of course important to provide some indication of the likely impact of duration of exposure to city life, as emphasised by the 2nd referee. You also indicate that data will be available from dryad, and this is indeed conditional upon processing and acceptance of your manuscript further. Please provide additional details in your covering letter.

Finally, while many of the key criteria for consideration of Evidence Synthesis articles have been met, I would ask you to look at the criteria below, and provide a very brief response, of how these have already been addressed or any modifications made. You will see below that you will be requested to upload a full response letter, and to facilitate the cross-referencing of changes incorporated, please indicate approximately where in your manuscript, major changes to the text have been incorporated. You will also find suggestions for inclusion of additional literature.

Importantly also, when putting the final touches to the article, please ensure wherever possible, that where relevant, you have addressed some of the questions below, that characterises the Evidence Synthesis article type, though I fully recognise, that many questions will not only partially apply to your manuscript (in your case, the following are especially pertinent: 1,2,3,4,6,7, 8,9,10. Such consideration is important, such that your article diverges clearly from the conventional type of review published elsewhere in PRSB.

1. Is the key policy-related question(s) articulated clearly?
2. Is there a clear justification in support of policy relevance?
3. Is the likely target audience identified clearly?
4. Does the search for literature utilise a comprehensive range of sources?
5. Does the synthesis article apply clearly documented inclusion criteria to all potentially relevant studies found during the search?
6. Is a clear methodology described to avoid bias?
7. Is your study objectively weighted according to methodological quality of cited literature?
8. Are knowledge gaps and priorities clearly identified?
9. Are outcomes/recommendations tangible in terms of likely impact?
10. Are all necessary supporting information available and accessible??

Thank you in advance for bringing this information together, and we look forward to receiving the revised new manuscript in due course.

Reviewer(s)' Comments to Author:

Referee: 1

Comments to the Author(s)

Dear authors,

thank you for contributing this well written and important paper. The topic has been well researched in the last 20 years, but only reviews and opinion pieces have been published that tried to summarise the existing evidence, so I really think that a meta-analysis is now timely.

I have no major comments on the procedures you have used for your meta-analysis, although I am not an expert of these methods so I am hoping that the other referees will be able to comment more in detail on such aspects.

My only major comment refers to the definition (or better, lack of definition) of what "urban" is in your analysis. I agree with you that species-specific responses to urbanisation are likely to drive much of the variation in the data available on this topic, as your results clearly show. However, I also strongly think that another reason where this variation might arise from is the great heterogeneity of the urban environment, as well as of the difference between urban and natural environments where the animals part of these studies were measured. You sort of recognise this aspect in your discussion (L255-257), but I think there is a scope to actually quantify such heterogeneity in the dataset and account for it in your models. For instance, it should be straightforward to obtain information about where the animals were sampled (exact coordinates) from the methods section of each paper, and then use such coordinates to quantify the level of urbanisation and/or "greenness" at each of these sites, for instance using published software (http://evolecol.hu/wp-content/uploads/2016/05/Seress_etal_2014_LUP.pdf). I think this will add a lot to your paper by greatly enhancing the interpretation of the results, thereby increasing the impact of your study. I don't think it will take a lot of time to do so since we are only talking about 34 results (68 estimates of urban/rural, which should be pretty fast to obtain using a software).

Another comment refers to when such measurements are taken. You distinguish this by using the variable "season" in your dataset. However, urbanised populations might differ in when they breed compared to rural counterparts, up to even a month (Partecke et al 2004 Proc B, Dominoni et al 2013 Proc B). Since we know that cort can vary on a seasonal basis, these samples might have been taken at different points of such annual cycle for urban and rural populations. There is nothing you can do about this now, but perhaps you could write one or two sentences in the discussion about it? Similarly, there is also a diel cycle in cort, but urban and rural animals can also differ in how they partition their activity between day and night along urban gradients, for instance mammals become more nocturnal when human activity is high (Gaynor et al 2017 Science). Again, if animals are sampled at the same time between urban and rural areas, these samples might have been taken at different times of their diel cycle in cort. So this could also be recognised in your discussion.

Thanks and I hope these comments will be helpful.

Referee: 2

Comments to the Author(s)

See attached file.

Author's Response to Decision Letter for (RSPB-2020-1754.R0)

See Appendix B.

RSPB-2020-1754.R1 (Revision)

Review form: Reviewer 1

Recommendation

Accept as is

Scientific importance: Is the manuscript an original and important contribution to its field?
Good

General interest: Is the paper of sufficient general interest?
Excellent

Quality of the paper: Is the overall quality of the paper suitable?
Good

Is the length of the paper justified?
Yes

Should the paper be seen by a specialist statistical reviewer?
Yes

Do you have any concerns about statistical analyses in this paper? If so, please specify them explicitly in your report.
No

It is a condition of publication that authors make their supporting data, code and materials available - either as supplementary material or hosted in an external repository. Please rate, if applicable, the supporting data on the following criteria.

Is it accessible?
Yes

Is it clear?
Yes

Is it adequate?
Yes

Do you have any ethical concerns with this paper?
No

Comments to the Author

The authors did a good job revising the paper based on my feedback. I have no further comments.

Review form: Reviewer 2 (Suzanne Mills)

Recommendation
Accept as is

Scientific importance: Is the manuscript an original and important contribution to its field?
Excellent

General interest: Is the paper of sufficient general interest?
Excellent

Quality of the paper: Is the overall quality of the paper suitable?
Excellent

Is the length of the paper justified?

Yes

Should the paper be seen by a specialist statistical reviewer?

No

Do you have any concerns about statistical analyses in this paper? If so, please specify them explicitly in your report.

No

It is a condition of publication that authors make their supporting data, code and materials available - either as supplementary material or hosted in an external repository. Please rate, if applicable, the supporting data on the following criteria.

Is it accessible?

No

Is it clear?

N/A

Is it adequate?

N/A

Do you have any ethical concerns with this paper?

No

Comments to the Author

Excellent job in replying to all the comments, carrying out additional analyses and in modifying the manuscript.

I only have 3 minor suggestions (one of which confused you in my first review!), which you may decide to take into consideration or not.

Line 130-132: I suggest adding "current levels" to the sentence either where I suggest or at the end: Baseline glucocorticoid levels were considered as those measured from cumulative sources (e.g. feather, hair) or current levels taken within 3 minutes of the capture of animals for blood. I believe this will help the general reader in distinguishing between cumulative and current baseline levels of GCs.

Lines 137-140: Does the paper report a time frame? If so can it be included here?

Line 253-255: I am sorry that my first comment confused the authors! What I tried to say was that while data was collected from 34 studies for baseline GCs, data was collected from only 19 studies for stress-induced GCs, and this is shown in the pie-charts in Fig. 1 on the left. Therefore I tried to suggest to add "from 19 studies" to this sentence, because as is, it reads that data for stress-induced GCs were also taken from 34 studies.

Decision letter (RSPB-2020-1754.R1)

10-Sep-2020

Dear Miss Iglesias-Carrasco

I am pleased to inform you that your Review manuscript RSPB-2020-1754.R1 entitled "Stress in the city: meta-analysis indicates no overall evidence for stress in urban vertebrates" has been accepted for publication in Proceedings B.

The referee(s) do not recommend any further changes. Therefore, please proof-read your manuscript carefully and upload your final files for publication. Because the schedule for publication is very tight, it is a condition of publication that you submit the revised version of your manuscript within 7 days. If you do not think you will be able to meet this date please let me know immediately.

To upload your manuscript, log into <http://mc.manuscriptcentral.com/prsb> and enter your Author Centre, where you will find your manuscript title listed under "Manuscripts with Decisions." Under "Actions," click on "Create a Revision." Your manuscript number has been appended to denote a revision.

You will be unable to make your revisions on the originally submitted version of the manuscript. Instead, upload a new version through your Author Centre.

1) A text file of the manuscript (doc, txt, rtf or tex), including the references, tables (including captions) and figure captions. Please remove any tracked changes from the text before submission. PDF files are not an accepted format for the "Main Document".

2) A separate electronic file of each figure (tiff, EPS or print-quality PDF preferred). The format should be produced directly from original creation package, or original software format. Please note that PowerPoint files are not accepted.

3) Electronic supplementary material: this should be contained in a separate file from the main text and the file name should contain the author's name and journal name, e.g. `authorname_procb_ESM_figures.pdf`

All supplementary materials accompanying an accepted article will be treated as in their final form. They will be published alongside the paper on the journal website and posted on the online figshare repository. Files on figshare will be made available approximately one week before the accompanying article so that the supplementary material can be attributed a unique DOI. Please see: <https://royalsociety.org/journals/authors/author-guidelines/>

4) Data-Sharing and data citation

It is a condition of publication that data supporting your paper are made available. Data should be made available either in the electronic supplementary material or through an appropriate repository. Details of how to access data should be included in your paper. Please see <https://royalsociety.org/journals/ethics-policies/data-sharing-mining/> for more details.

<http://datadryad.org/submit?journalID=RSPB&manu=RSPB-2020-1754.R1> which will take you to your unique entry in the Dryad repository.

Once again, thank you for submitting your manuscript to Proceedings B and I look forward to receiving your final version. If you have any questions at all, please do not hesitate to get in touch.

Sincerely,
Professor Gary Carvalho
Editor, Proceedings B
mailto:proceedingsb@royalsociety.org

Comments to Author:

Thank you for the comprehensive revision and constructive responses to all points raised. As you will see, together with the 2 referees, we are now happy to proceed to the final stage of consideration for your manuscript. There are a few remaining, relatively minor issues raised by one of the referees, that I would ask you to address. There is no need for the manuscript to be seen again by the referees or editorial team. I also appreciate the very detailed and helpful responses to my particular questions concerning criteria relating to the evidence synthesis article format. All responses are clear, and I am happy to now proceed. Thank you once again for your interest in publishing your manuscript as an Evidence Synthesis article in PRSB. We look forward to seeing the remaining minor revisions, and taking your interesting and insightful manuscript forward for publication.

Reviewer(s)' Comments to Author:

Referee: 1

Comments to the Author(s)

The authors did a good job revising the paper based on my feedback. I have no further comments.

Referee: 2

Comments to the Author(s)

Excellent job in replying to all the comments, carrying out additional analyses and in modifying the manuscript.

I only have 3 minor suggestions (one of which confused you in my first review!), which you may decide to take into consideration or not.

Line 130-132: I suggest adding “current levels” to the sentence either where I suggest or at the end: Baseline glucocorticoid levels were considered as those measured from cumulative sources (e.g. feather, hair) or current levels taken within 3 minutes of the capture of animals for blood. I believe this will help the general reader in distinguishing between cumulative and current baseline levels of GCs.

Lines 137-140: Does the paper report a time frame? If so can it be included here?

Line 253-255: I am sorry that my first comment confused the authors! What I tried to say was that while data was collected from 34 studies for baseline GCs, data was collected from only 19 studies for stress-induced GCs, and this is shown in the pie-charts in Fig. 1 on the left. Therefore I tried to suggest to add “from 19 studies” to this sentence, because as is, it reads that data for stress-induced GCs were also taken from 34 studies.

Decision letter (RSPB-2020-1754.R2)

14-Sep-2020

Dear Miss Iglesias-Carrasco

I am pleased to inform you that your manuscript entitled "Stress in the city: meta-analysis indicates no overall evidence for stress in urban vertebrates" has been accepted for publication in Proceedings B.

Your article has been estimated as being 11 pages long. Our Production Office will be able to confirm the exact length at proof stage.

Open Access

Paper charges

Sincerely,

Proceedings B

Appendix A

Stress in the city: meta-analysis indicates no overall evidence of higher levels of stress hormones in urban vertebrates

The authors provide an interesting meta-analysis comparing baseline and stress-induced responses of animals living in both cities and rural areas. Furthermore, the authors partition the data between the method of glucocorticoid sampling (feather/fur versus blood) and also sex, reproductive season and taxon. The data selected from the literature is exhaustive and the analyses appropriate and in-depth. The lack of differences in glucocorticoid levels between cities and rural areas is well discussed and explanations are given. The paper is well written and the figures clear, appealing and self explanatory.

General comment:

The authors have slightly simplified the response of the neuroendocrine system to a stressor by only referring to the case of chronic stress (lines 48-58). The authors also suggest that stress-induced responses are expected to be elevated in previously chronically stressed individuals (lines 50-51). However, the neuroendocrine system can also decrease GC secretion in chronically stressed individuals resulting in reduced baseline concentrations (Rich and Romero, 2005; Cyr and Romero, 2007; but also Herbert et al., 2006)

- Herbert, J., Goodyer, I.M., Grossman, A.B., Hastings, M.H., de Kloet, E.R., Lightman, S.L., Lupien, S.J., Roozendaal, B., Seckl, J.R., 2006. Do corticosteroids damage the brain? *J. Neuroendocrinol.* 18, 393–411

Furthermore, chronic stress may also compromise the HPA axis with the opposite impact of stress-induced responses, i.e. their downregulation. The examples I give here are unfortunately all from fish and not in urban environments, but I include them anyway:

- Hontela, A., Rasmussen, J.B., Audet, C., G C, 1992. Impaired cortisol stress response in fish from environments polluted by PAHs, PCBs and mercury. *Arch. Environ. Contam. Toxicol.* 22, 278-283.
- Norris, D.O., Donahue, S., Dores, R.M., Lee, J.K., Maldonado, T.A., Ruth, T., Woodling, J.D., 1999. Impaired adrenocortical response to stress by brown trout, *Salmo trutta*, living in metal-contaminated waters of the Eagle River, Colorado. *Gen. Comp. Endocrinol. Contam. Toxicol.* 22, 278-283.
- Pottinger, T.G., Henrys, P.A., Williams, R.J., Matthiesson, P., 2013. The stress response of three-spined sticklebacks is modified in proportion to effluent exposure downstream of wastewater treatment works. *Aquat. Toxicol.* 126, 382-392.
- Koakoski, G., Quevedo, R.M., Ferreira, D., et al., 2014. Agrichemicals chronically

inhibit the cortisol response to stress in fish. *Chemosphere* 112, 85-91.

- Mills, S.C., Beldade, R., Henry, L., Laverty, D., Nedelec, S.L., Simpson, S.D., & Radford, A.N. 2020. Hormonal and behavioural effects of motorboat noise on wild coral reef fish. *Environmental Pollution* 262: 114250

Furthermore, baseline and stress-induced responses will vary depending on whether the individuals developed/were raised within cities or only migrated there as adults. Stressful conditions during development may permanently modify the development of the HPA axis - in some cases increasing the stress-induced response in later life (e.g. Beauguard et al 2019). Whereas, stressful conditions only during the adult stage may either increase or decrease baseline and stress-induced response due to plasticity in the HPA axis. Therefore, the response of the endocrine system is complex.

Could the authors include a sentence on whether the animals used in this study were urban-raised or not? Would this be interesting to add as a moderator?

Lines 63-66: Again it is a simplification to state that the prediction is for urban populations to have higher levels of GCs due to chronic stress without stating whether this refers to baseline or stress-induced levels as the predictions are multiple and may be opposing. Furthermore, the authors go on to state that “some studies find higher glucocorticoid levels in urban populations (23, 24), others find no effects (25, 26) or even the opposite effect (27, 28)” without stating whether this refers to baseline or stress-induced GCs and hence this is misleading. For example, the authors refer to reference 26 as no effect – this is true for baseline levels, whereas stress-induced GCs were lower from urban areas. I suggest making clearer predictions for each type of stress response.

What is clear is that the responses of the HPA axis to stressors are varied.

I only have one caveat for the paper. I find the comparison between hair/feather and feces and blood very interesting. As the authors point out hair and feathers represent GCs over the previous weeks to months, however, during this period individuals may have faced different scenarios: migrated between cities and rural areas; remained in the cities the entire period, only recently arrived in the city days prior to capture – we do not know unless the individuals had been tagged. If individuals are migrating between urban and non-urban areas, or only recently arrived there, then the urban impact on feather/fur GC levels will be low and bias the results. However, there is no discussion of this, or whether such data can be included as a moderator. Feces represent GC levels accumulated over hours or days and so individuals are less likely to have migrated between cities and natural areas. However, blood represents GC levels at capture and therefore we can be certain that blood GCs accurately represent the stress levels of individuals to the site in which they were found. Certainly, blood GC may be confounded by handling stress, however, as long as time from capture to blood flowing in the syringe has been

recorded, was below 3-4 minutes and showed no difference between urban and rural areas, then this should not influence mean effects. It is also interesting that out of all the results in Figure 2, blood baseline GCs and blood stress-induced GCs, especially Free, showed the highest effect size and were higher in urban areas. I understand that stress-induced GCs were only measured from blood samples and no differences were found between urban and rural areas. However, could the analysis be repeated on baseline GCs but only including the blood (both total and free) samples so that any bias in individual movement can be removed from the analysis? I would be interested to see the result.

Other general comments:

Lines 136-138: Please explain in more detail the reasons behind the use of the final measurement of stress-induced glucocorticoid levels when multiple measures were taken. Following a stressor glucocorticoid levels increase and then fall, therefore, I would suggest using the maximum level rather than the final. Also please reference these papers that used multiple measures so the author can easily refer to them.

Lines 293-300: An interesting line of discussion could be to draw a similarity between urban living and invasive species. Interesting work has compared personality and GC levels of individuals at the edge of their species range (invaders) and those at the range core. One could envisage that city dwellers are infact colonisers of this new habitat and hence share traits with invasive individuals. It would be further interesting to determine whether the success of individuals that were transplanted into cities was related to their different levels of stress response and hence determine whether the lack of differences found were simply due to phenotypic plasticity, but initially there was natural selection on individuals with certain stress responses. See Martin et al Gen Comp Endo 2017 for further reading.

- Martin, L.B., et al., 2017. Corticosterone regulation in house sparrows invading Senegal. *General and Comparative Endocrinology* 250: 15-20

Specific points:

Methods: please give more detail of how stress-induced measures were taken – I assume in response to capture and restraint but this is not stated. Mean and/or range of time from capture to blood sampling could also be given.

Line 51: a stress-induced response refers to the level of glucocorticoids in response to an environmental stressor or a standardized stressor (e.g. capture and restraint). The current wording suggests that the authors are confusing this with an excessive glucocorticoid surge in response to an acute stressor in chronically stressed individuals. This may only be unfortunate timing of when the term “stress-induced response” is first introduced, but I suggest that the authors introduce the term “stress-induced response” on line 43 instead which will remove any confusion.

Line 61, and 189 and 191 and 214, Table 1 throughout, Figure 1 b) title: change to stress-induced (add hyphen to be consistent with rest of manuscript)

Lines 68-69: I suggest ordering feather and blood to their corresponding levels of stress i.e. cumulative and current and using respectively for clarity.

Line 105: change glucocorticoids to glucocorticoid (singular)

Line 140: what are moderators? I understand this to be sex, lifestage, season and source of hormone. I would suggest introducing this term therefore on Lines 121 when these moderators are first introduced.

Lines 139-140: Baseline levels (not stress-induced levels) for juveniles were excluded even though the sample size (species and study) were > 2 and effect size was > 6 , so I suggest that these values be modified to include the exclusion criteria for excluding the level of juveniles. However, I fear by increasing the exclusion criteria for juveniles this will mean that other levels will need to be excluded. Alternatively, keep the exclusion criteria as stated and include juveniles in the model.

Line 157: first quotation mark is incorrect.

Line 189 and 230: change to stress-induced (not stressed)

Line 214: As Figure 1 first shows the distribution of studies across taxa and that the number of studies (34) is explained for baseline levels but not for stress-induced, I suggest adding 19 studies e.g. "... and from the 19 studies there were 55 effects from 15 species for stress-induced glucocorticoids".

Line 218: mean g not the same as in Table S2: 0.124 in text but 0.105 in Table.

Line 219: mean g not the same as in Table S3: 0.065 in text but 0.068 in Table.

Line 221: CI values not the same as in Table S2: 0.213 in text but 0.222 in Table.

Line 231: mean g not the same as in Table S4: 0.041 in text but -0.016 in Table.

Line 232: mean g not the same as in Table S4: 0.029 in text but -0.009 in Table.

There is no reference to Figure 2 in the whole text! I suggest adding it to lines 225-228.

Line 352: change advise to advice. Also capitalise We thank.

Figure 2 legend, line 617: change in to on

Table S2: I would suggest changing basal to baseline in the table legend.

Table S4: change to stress-induced (add hyphen to be consistent)

Appendix B

Dear Editor,

Thank you for considering a revised version of our manuscript now entitled “Stress in the city: meta-analysis indicates no overall evidence for stress in urban vertebrates” for publication in Proceedings of the Royal Society B. This revision addresses the comments by the Associated editor and two reviewers.

Below we provide a point-by-point list of the actions we took to address the comments.

Sincerely,

Maidier Iglesias-Carrasco on behalf of all authors.

Associate Editor

Board Member: 1

Comments to Author:

Thank you for submitting your manuscript for consideration as an Evidence Synthesis article in PRSB. Your work has now been seen by 2 experts in the field, and I have read the manuscript myself. I am pleased to say that there is a consensus that your article is interesting, highly pertinent for evidence synthesis, and provides a considered and objective overview of key literature and ideas. As such, I encourage you to revise the manuscript, in line with the constructive and detailed suggestions below. Overall (though please do view the criteria reproduced below) I can say that I am satisfied with the level of transparency and statistical rigour in your meta-analysis, especially the explicit inclusion of criteria for choice of data/literature, and the associated statistical and sensitivity analyses. Among the various key issues raised, is the need to provide a clearer and standardised definition of what constitutes an urban environment, and I would encourage you to consider inclusion of spatial coordinates, and the referee proposes some appropriate software, to provide a quantitative comparator of specific sample sites. I appreciate that this may require an additional workload, but my view is that it will provide a more informed and consistent classification in relation to the inferences drawn. I also emphasise the need to provide some consideration of the likely impact of when samples were collected, and whether any seasonal variation in reproductive cycles for example, are likely to have an impact. There is also a need to provide a more explicit, though brief, consideration of the complexity of stress-induced endocrine responses, as well as a more considered approach of the likely impacts of using a diversity of sample types, encompassing feathers, hair, blood and faeces. It is also of course important to provide some indication of the likely impact of duration of exposure to city life, as emphasised by the 2nd referee. You also indicate that data will be available from Dryad, and this is indeed conditional upon processing and acceptance of your manuscript further. Please provide additional details in your covering letter.

We thank the Associated Editor for their positive assessment of our study.

As suggested by the editor we have addressed all the issues raised by the two reviewers. Specifically, in response to Reviewer 1 we have now added definitions of urban and non-urban environments, which is based on the criteria used in the original studies. We have also re-examined the 34 studies included in our meta-analysis to search for the specific coordinates of the sampling sites. However, only 16 of the 34 studies provided this information, which hampers our ability to include an urbanization score as a moderator in our analysis. Instead, we have therefore collected data on the population size of each city used in the meta-analysis to include it as a moderator that might explain some of the variation among cities. Please note that there was no relationship between city population size and the effect of urbanisation on stress responses. Including city size as a moderator did not alter our previous results/conclusions. We have also acknowledged in our Discussion the potential impact that phenological differences between urban and non-urban populations might have on our results.

In response to reviewer 2 we have now reworded our Introduction and Discussion to provide further information about the complexity of the stress response, and how the predictions for both baseline and stress-induced responses might depend on urbanisation. We have also extended our Discussion to explain the likely effects of using cumulative vs current hormone sampling methods and then related these to the duration of exposure to city life. In addition, we have reanalysed our data to include all the suggestions made by Reviewer 2.

We have now published our dataset in Dryad, doi:10.506/dryad.pzgmsbcj5.

Finally, while many of the key criteria for consideration of Evidence Synthesis articles have been met, I would ask you to look at the criteria below, and provide a very brief response, of how these have already been addressed or any modifications made. You will see below that you will be requested to upload a full response letter, and to facilitate the cross-referencing of changes incorporated, please indicate approximately where in your manuscript, major changes to the text have been incorporated. You will also find suggestions for inclusion of additional literature.

Importantly also, when putting the final touches to the article, please ensure wherever possible, that where relevant, you have addressed some of the questions below, that characterises the Evidence Synthesis article type, though I fully recognise, that many questions will not only partially apply to your manuscript (in your case, the following are especially pertinent: 1,2,3,4,6,7, 8,9,10. Such consideration is important, such that your article diverges clearly from

the conventional type of review published elsewhere in PRSB.

1. Is the key policy-related question(s) articulated clearly?

The question we ask is whether urban areas are stressful for wildlife (line 12, and lines 40-41). While the effects of urbanisation on wildlife are of interest to urban planners and conservation managers, the field is young and it is difficult to make recommendations that could be explicitly used by policy makers. Instead we make recommendations to improve the quality of future research (see Conclusions), which will, in time, improve the value of this research for policy makers.

2. Is there a clear justification in support of policy relevance?

Urban areas continue to grow in size and density, and the public are increasingly concerned about the effects of urbanisation on wildlife (stated in line 12, lines 35-38). Understanding the stress response of animals to urbanisation could provide early indicators of population health, as well as information about the physiological responses of individuals to urban environments. Results supporting or refuting tests that urbanisation induces stress are currently mixed, and a meta-analysis is therefore timely and relevant (stated in lines 14 and 83-91).

3. Is the likely target audience identified clearly?

The target audience is anyone generally interested in ecology, which is a key reason why we have kept the manuscript fairly broad and providing several explanations for our results. More specifically, our study will be of interest to urban ecologist and physiologist, so we also provide more specific information targeted at these researchers. The title covers all these topics.

4. Does the search for literature utilise a comprehensive range of sources?

As stated in Line 97 in the Methods, we used Web of Science and Scopus to perform our literature search. These two sources provide a comprehensive and unbiased account of most of the relevant scientific literature.

5. Does the synthesis article apply clearly documented inclusion criteria to all potentially relevant studies found during the search?

We provide clear reasons for the exclusion and inclusion of the studies found during the search (Lines 105-123) and Prisma diagram in Supplementary Materials), as well as information about the exclusion criteria for each of the effect sizes included/excluded (Line 176 and Table S1).

6. Is a clear methodology described to avoid bias?

We have provided information about the search methods, searching criteria and exclusion criteria used at each step of the process. For further clarity we have also provided a table in the Supplementary Information (S1) about the specific exclusion criteria for the effect sizes in each model. A reader can therefore replicate our models. Also, please note that our modelling approach was decided prior to the data analysis to avoid any bias.

7. Is your study objectively weighted according to methodological quality of cited literature?

In the fields of ecology and evolution the robustness of a study is usually equivalent to its sample size. Larger sample sizes provide more robust results. The methods associated with calculating effect sizes in our meta-analysis are strictly weighted according to the studies' sample size. We do not provide a subjective assessment of the quality of each study as this is difficult in our discipline, and we believe that it has the potential to introduce bias.

8. Are knowledge gaps and priorities clearly identified?

We clearly state in the Introduction (Lines 14, 80) that there is a lack of a formal synthesis of the effects of urbanization on wildlife stress responses. Throughout the Discussion, and especially in the Conclusions, we provide recommendations on future research on the topic (Lines 431-445). By following these recommendations we will improve our understanding of how wildlife cope with urban habitats to better inform policy-making decisions in the future.

9. Are outcomes/recommendations tangible in terms of likely impact?

We have given extensive explanations and recommendations about how to improve data collection in future studies to understand the effect of urbanization on wildlife. These recommendations are of critical importance as urban environments continue to increase in size. The stress response of animals to urbanisation could provide early indicators of population health and the ability to cope with the novel urban habitat.

10. Are all necessary supporting information available and accessible??

We have included the results from the most relevant sensitivity analysis in the Supplementary Information, along with all parameter estimates from our results. The Supplementary Information also contains the funnel plots used to check for publication bias. All supplementary information is provided with the paper and the raw data collected from the primary studies are provided in Dryad doi:10.506/dryad.pzgmsbcj5.

Thank you in advance for bringing this information together, and we look forward to receiving the revised new manuscript in due course.

Reviewer(s)' Comments to Author:

Referee: 1

Comments to the Author(s)

Dear authors,

thank you for contributing this well written and important paper. The topic has been well researched in the last 20 years, but only reviews and opinion pieces have been published that tried to summarise the existing evidence, so I really think that a meta-analysis is now timely.

I have no major comments on the procedures you have used for your meta-analysis, although I am not an expert of these methods so I am hoping that the other referees will be able to comment more in detail on such aspects.

We thank the reviewer for their positive assessment of our study.

My only major comment refers to the definition (or better, lack of definition) of what "urban" is in your analysis. I agree with you that species-specific responses to urbanisation are likely to drive much of the variation in the data available on this topic, as your results clearly show. However, I also strongly think that another reason where this variation might arise from is the great heterogeneity of the urban environment, as well as of the difference between urban and natural environments where the animals part of these studies were measured. You sort of recognise this aspect in your discussion (L255-257), but I think there is a scope to actually quantify such heterogeneity in the dataset and account for it in your models. For instance, it should be straightforward to obtain information about where the animals were sampled (exact coordinates) from the methods section of each paper, and then use such coordinates to quantify the level of urbanisation and/or "greenness" at each of these sites, for instance using published software (http://evolecol.hu/wp-content/uploads/2016/05/Seress_etal_2014_LUP.pdf). I think this will add a lot to your paper by greatly enhancing the interpretation of the results, thereby increasing the impact of your study. I don't think it will take a lot of time to do so since we are only talking about 34 results (68 estimates of urban/rural, which should be pretty fast to obtain using a software).

We agree with the reviewer that differences between urban and non-urban sites across the studies might partly explain the lack of a general effect in our meta-analysis. We liked the reviewer's suggestion, to collect co-ordinate data from the papers and to include an "urbanization score" in our analysis to reduce the potential noise created by this source of variation across studies. However, to our surprise, only half of the 34 studies that contributed data to our meta-analysis provided the exact coordinates for their study sites. In addition, some of the studies collected animals from several sites within the same urban area, which made measuring a precise urbanization score more difficult than expected. Further, it is difficult to know whether the maps available today reflect the greenness and urbanization at the time of data collection.

Consequently, instead of using the method suggested by the reviewer we have attempted to control for the "degree of urbanisation" by collecting data on the population size of each city studied and using this as a moderator in our meta-analysis. Although we know this method is also imperfect and that the stress response might differ between specific areas within a city, we still expect that the population size of a city offers a broad scale indication of the level of urbanisation. Of course, our meta-analysis is designed to test for general differences between habitats, with the assumption that paired differences between urban and non-urban sites will be stronger than those within urban and non-urban habitats. We hope that our study will guide further research, and in so doing improve data collection and ensure better future reporting which will greatly enhance the ability of meta-analysts to explore more complex and detailed questions such as those the reviewers propose.

Our new analyses that control for city population size show that there was no effect of it as a moderator on the effect sizes (see new Figure S3). City size did not affect the direction or strength of the observed effect sizes. We have now added in our Methods how we define "urban" and "non-urban" habitats (Lines 141-149). We have also added the new models that include population size as moderator (Lines 158, 214, 223, 231, 267, 275). Finally, we have included the results from the new model in Tables S3 and S4, and covered these models in our revised Discussion (Line 299).

Another comment refers to when such measurements are taken. You distinguish this by using the variable "season" in your dataset. However, urbanised populations might differ in when they breed compared to rural counterparts, up to even a month (Partecke et al 2004 Proc B, Dominoni et al 2013 Proc B). Since we know that cort can vary on a a seasonal basis, these samples might have been taken at different points of such annual cycle for urban and rural populations. There is nothing you can do about this now, but perhaps you could write one or two sentences in the discussion about it? Similarly, there is also a diel cycle in cort, but urban and rural animals can also differ in how they partition their actiity between day and night along urban gradients, for instance mammals become more nocturnal when human activity is high (Gaynor et al 2017 Science). Again, if animals are sampled at the same time between urban and rural areas, these samples might have been taken at different times of their diel cycle in cort. So this could also be recognised in your discussion.

This is a great point and one that we had previously considered incorporating into our analysis. However, the studies from which our data was collected do not report or take into account phenological differences between urban and non-urban sites. Although interesting, investigation of this potential source of variation between populations is very difficult as considerable knowledge about the populations being studied is needed. The extent of variation in phenology between urban and non-urban populations is likely to differ between species, cities and possibly even years. The available studies that we used controlled for potential temporal variation in sampling by sampling urban and non-urban areas at the same times of year. This is also a legitimate approach, but it makes it impossible to incorporate the reviewer's point into our analysis. However, we agree that this is a very interesting idea that could influence our results, and should be incorporated into future experimental studies. We have now highlighted this issue in our Discussion (Line 318).

Referee 2.

The authors provide an interesting meta-analysis comparing baseline and stress-induced responses of animals living in both cities and rural areas. Furthermore, the authors partition the data between the method of glucocorticoid sampling (feather/fur versus blood) and also sex, reproductive season and taxon. The data selected from the literature is exhaustive and the analyses appropriate and in-depth. The lack of differences in glucocorticoid levels between cities and rural areas is well discussed and explanations are given. The paper is well written and the figures clear, appealing and self explanatory.

We thank the reviewer for their general positive assessment of our meta-analysis.

General comment:

The authors have slightly simplified the response of the neuroendocrine system to a stressor by only referring to the case of chronic stress (lines 48-58). The authors also suggest that stress-induced responses are expected to be elevated in previously chronically stressed individuals (lines 50-51). However, the neuroendocrine system can also decrease GC secretion in chronically stressed individuals resulting in reduced baseline concentrations (Rich and Romero, 2005; Cyr and Romero, 2007; but also Herbert et al., 2006)

• Herbert, J., Goodyer, I.M., Grossman, A.B., Hastings, M.H., de Kloet, E.R., Lightman, S.L., Lupien, S.J., Roozendaal, B., Seckl, J.R., 2006. Do corticosteroids damage the brain? *J. Neuroendocrinol.* 18, 393–411

Furthermore, chronic stress may also compromise the HPA axis with the opposite impact of stress-induced responses, i.e. their downregulation. The examples I give here are unfortunately all from fish and not in urban environments, but I include them anyway:

- Hontela, A., Rasmussen, J.B., Audet, C., G C, 1992. Impaired cortisol stress response in fish from environments polluted by PAHs, PCBs and mercury. *Arch. Environ. Contam. Toxicol.* 22, 278-283.
- Norris, D.O., Donahue, S., Dores, R.M., Lee, J.K., Maldonado, T.A., Ruth, T., Woodling, J.D., 1999. Impaired adrenocortical response to stress by brown trout, *Salmo trutta*, living in metal-contaminated waters of the Eagle River, Colorado. *Gen. Comp. Endocrinol. Contam. Toxicol.* 22, 278-283.
- Pottinger, T.G., Henrys, P.A., Williams, R.J., Matthiesson, P., 2013. The stress response of three-spined sticklebacks is modified in proportion to effluent exposure downstream of wastewater treatment works. *Aquat. Toxicol.* 126, 382-392.
- Koakoski, G., Quevedo, R.M., Ferreira, D., et al., 2014. Agrichemicals chronically 1 inhibit the cortisol response to stress in fish. *Chemosphere* 112, 85-91.
- Mills, S.C., Beldade, R., Henry, L., Laverty, D., Nedelec, S.L., Simpson, S.D., & Radford, A.N. 2020. Hormonal and behavioural effects of motorboat noise on wild coral reef fish. *Environmental Pollution* 262: 114250

We thank the reviewer for highlighting that we oversimplified our explanation of the neuroendocrine system and how animals might respond to stress associated with living in urban environments. Our intention had been to keep our explanation as simple as possible to make our study accessible to non-physiologists. However, in hindsight we agree that by only mentioning that the main expectation was for baseline and stress-induced responses to increase in urban areas, we downplayed other potential effects of urbanisation on stress responses. We have now rewritten part of our Introduction (Lines 48-57) and included extra information in our Discussion (Lines 345-359) to better reflect the potential complexity of any stress-response. We have also made small changes throughout the text to match the new explanations (e.g., title, line 16, line 286).

Furthermore, baseline and stress-induced responses will vary depending on whether the individuals developed/were raised within cities or only migrated there as adults. Stressful conditions during development may permanently modify the development of the HPA axis - in some cases increasing the stress-induced response in later life (e.g. Beaugard et al 2019). Whereas, stressful conditions only during the adult stage may either increase or decrease baseline and stress-induced response due to plasticity in the HPA axis. Therefore, the response of the endocrine system is complex.

Could the authors include a sentence on whether the animals used in this study were urban-raised or not? Would this be interesting to add as a moderator?

We thank the reviewer for their suggestion. Unfortunately, the relevant information is not given in the available studies, which hampers our ability to include the origin of individuals as a moderator. However, we now discuss this topic in our Discussion (Lines 378-402) and point out that it is a profitable line of future inquiry.

Lines 63-66: Again it is a simplification to state that the prediction is for urban populations to have higher levels of GCs due to chronic stress without stating whether this refers to baseline or stress-induced levels as the predictions are multiple and may be opposing. Furthermore, the authors go on to state that “some studies find higher glucocorticoid levels in urban populations (23, 24), others find no effects (25, 26) or even the opposite effect (27, 28)” without stating whether this refers to baseline or stress-induced GCs and hence this is misleading. For example, the authors refer to reference 26 as no effect – this is true for baseline levels, whereas stress-induced GCs were lower from urban areas. I suggest making clearer predictions for each type of stress response. What is clear is that the responses of the HPA axis to stressors are varied.

We thank the reviewer for their comment. We have now rewritten the relevant section of our Introduction (Lines 67-78) to make the separate predictions for baseline and stress-induced responses clear. We have then connected this section with another section that was modified in response to the reviewer’s previous comment (in Lines 48-57) where we acknowledge the complexity of the HPA response, and the difficulty of making directional predictions. We have also added extra information about this issue to our Discussion (Lines 345-359).

I only have one caveat for the paper. I find the comparison between hair/feather and feces and blood very interesting. As the authors point out hair and feathers represent GCs over the previous weeks to months, however, during this period individuals may have faced different scenarios: migrated between cities and rural areas; remained in the cities the entire period, only recently arrived in the city days prior to capture – we do not know unless the individuals had been tagged. If individuals are migrating between urban and non-urban areas, or only recently arrived there, then the urban impact on feather/fur GC levels will be low and bias the results. However, there is no discussion of this, or whether such data can be included as a moderator. Feces represent GC levels accumulated over hours or days and so individuals are less likely to have migrated between cities and natural areas. However, blood represents GC levels at capture and therefore we can be certain that blood GCs accurately represent the stress levels of individuals to the site in which they were found. Certainly, blood GC may be confounded by handling stress, however, as long as time from capture to blood flowing in the syringe has been recorded, was below 3-4 minutes and showed no difference between urban and rural areas, then this should not influence mean effects. It is also interesting that out of all the results in Figure 2, blood baseline GCs and blood stress-induced GCs, especially Free, showed the highest effect size and were higher in urban areas. I understand that stress-induced GCs were only measured from blood samples and no differences were found between urban and rural areas. However, could the analysis be repeated on baseline GCs but only including the blood (both total and free) samples so that any bias in individual movement can be removed from the analysis? I would be interested to see the result.

This is a very good point and something to take into account. We agree that any migration between urban/non urban habitats could affect the reliability of cumulative vs current measures of hormones. Unfortunately, there is no information in the original studies about movement patterns of the individuals captured. This could be because of the difficulties of obtaining this information (need to mark individuals and control for movement patterns) or because researchers simply assume that individuals are local residents. Given the lack of information we cannot control for this variable in our analysis. However, we have reanalysed the data using only blood glucocorticoids as suggested by the reviewer. The results remained the same, with no significant effects of urbanization on stress levels, suggesting that including the cumulative measures does not affect our results. We have added the new results in the Supplementary Information (Table S5) and discussed them in the Discussion, Lines 378-402.

Other general comments:

Lines 136-138: Please explain in more detail the reasons behind the use of the final measurement of stress-induced glucocorticoid levels when multiple measures were taken. Following a stressor glucocorticoid levels increase and then fall, therefore, I would suggest using the maximum level rather than the final. Also please reference these papers that used multiple measures so the author can easily refer to them.

We used the maximum as this was usually a more similar time frame to that used by other studies (e.g. 30 or 60 mins). However, we have checked the discarded measurements and, in a few cases, the peak of the stress response does not coincide with the final measurement selected. We have, therefore, conducted a reanalysis of the stress-induced response data using the *maximum* measurements. The results remain the same, with no significant effects of the habitat on the stress response of the animals (Line 171 and Supplementary Information Table S4).

Lines 293-300: An interesting line of discussion could be to draw a similarity between urban living and invasive species. Interesting work has compared personality and GC levels of individuals at the edge of their species range (invaders) and those at the range core. One could envisage that city dwellers are in fact colonisers of this new habitat and hence share traits with invasive individuals. It would be further interesting to determine whether the success of individuals that were transplanted into cities was related to their different levels of stress response and hence determine whether the lack of differences found were simply due to phenotypic plasticity, but initially there was

natural selection on individuals with certain stress responses. See Martin et al Gen Comp Endo 2017 for further reading.

• Martin, L.B., et al., 2017. Corticosterone regulation in house sparrows invading Senegal. *General and Comparative Endocrinology* 250: 15-20

This is a very nice point and one that we have not considered. We believe that adding this idea offers an interesting contrast to the general assumption that any differences in the stress response between urban and non-urban individuals arise from chronic stress. We have now introduced this idea in our Discussion (Lines 417-427).

Specific points:

Methods: please give more detail of how stress-induced measures were taken – I assume in response to capture and restraint but this is not stated. Mean and/or range of time from capture to blood sampling could also be given.

We have now added the definition of what was considered baseline and stress-induced levels of glucocorticoids in Lines 130-140.

Line 51: a stress-induced response refers to the level of glucocorticoids in response to an environmental stressor or a standardized stressor (e.g. capture and restraint). The current wording suggests that the authors are confusing this with an excessive glucocorticoid surge in response to an acute stressor in chronically stressed individuals. This may only be unfortunate timing of when the term “stress-induced response” is first introduced, but I suggest that the authors introduce the term “stress-induced response” on line 43 instead which will remove any confusion.

We thank the reviewer for their suggestion. We agree that where we introduced the term “stress-induced response” was not ideal, and have now moved it to Line 42.

Line 61, and 189 and 191 and 214, Table 1 throughout, Figure 1 b) title: change to stress-induced (add hyphen to be consistent with rest of manuscript)

Done, we have now added a hyphen to this term throughout the manuscript.

Lines 68-69: I suggest ordering feather and blood to their corresponding levels of stress i.e. cumulative and current and using respectively for clarity.

We have reordered these terms in the text (Line 79).

Line 105: change glucocorticoids to glucocorticoid (singular)

Done

Line 140: what are moderators? I understand this to be sex, lifestage, season and source of hormone. I would suggest introducing this term therefore on Lines 121 when these moderators are first introduced.

We have now introduced the term moderator earlier (Line 158) and explain that sex, life stage, season, source of hormone and human population size were used as moderators in the models.

Lines 139-140: Baseline levels (not stress-induced levels) for juveniles were excluded even though the sample size (species and study) were > 2 and effect size was > 6, so I suggest that these values be modified to include the exclusion criteria for excluding the level of juveniles. However, I fear by increasing the exclusion criteria for juveniles this will mean that other levels will need to be excluded. Alternatively, keep the exclusion criteria as stated and include juveniles in the model.

In our previous analysis we did not include juveniles because we wanted to keep our “all taxa” consistent with our analysis of birds only. However, we agree that juveniles pass the exclusion criteria in the dataset for baseline glucocorticoids and all taxa. We have therefore now included them in these analyses. We have also added a model where life stage is included as a moderator (see Lines 154, 175, 221). Note, however, that the results do not change and that life stage still does not affect effect sizes when we include it as a moderator. Table S3.

Line 157: first quotation mark is incorrect.

We have corrected this typo.

Line 189 and 230: change to stress-induced (not stressed)

We have corrected the typo

Line 214: As Figure 1 first shows the distribution of studies across taxa and that the number of studies (34) is explained for baseline levels but not for stress-induced, I suggest adding 19 studies e.g. "... and from the 19 studies there were 55 effects from 15 species for stress-induced glucocorticoids".

We have to admit that we are a little confused by this comment. We think the reviewer is saying that we have included sample sizes for baseline levels but not for stress induced, however, the sample sizes for the stress induced levels are shown in panel b of this figure. We have now added a description to the figure caption stating what is depicted in panel a) and panel b). We hope that this makes things clearer.

Line 218: mean g not the same as in Table S2: 0.124 in text but 0.105 in Table.

We have now updated the values in both the text and the Supplementary tables to match the new results.

Line 219: mean g not the same as in Table S3: 0.065 in text but 0.068 in Table.

We have now updated the values in both the text and the Supplementary tables to match the new results.

Line 221: CI values not the same as in Table S2: 0.213 in text but 0.222 in Table.

We have now updated the values in both the text and the Supplementary tables to match the new results.

Line 231: mean g not the same as in Table S4: 0.041 in text but -0.016 in Table.

We have now updated the values in both the text and the Supplementary tables to match the new results.

Line 232: mean g not the same as in Table S4: 0.029 in text but -0.009 in Table.

We have now updated the values in both the text and the Supplementary tables to match the new results.

There is no reference to Figure 2 in the whole text! I suggest adding it to lines 225- 228.

We now refer to Figure 2 in Lines 268 and 277.

Line 352: change advise to advice. Also capitalise We thank.

Done.

Figure 2 legend, line 617: change in to on

Done.

Table S2: I would suggest changing basal to baseline in the table legend.

We have changed it to baseline.

Table S4: change to stress-induced (add hyphen to be consistent)

We have reworded it accordingly.